# The impact of multiple abiotic stresses on *ns-LTP2.8* gene transcript and ns-LTP2.8 protein accumulation in germinating barley (*Hordeum vulgare* L.) embryos

**Michał Kempa, Krzysztof Mikołajczak, Piotr Ogrodowicz, Tomasz Pniewski, Paweł Krajewski, Anetta Kuczyńska** *

Institute of Plant Genetics, Polish Academy of Sciences, Poznan, Poland

* akuc@igr.poznan.pl

**Data Availability Statement:** All relevant data are within the manuscript and its Supporting Information files.

## Abstract

Abiotic stresses occur more often in combination than alone under regular field conditions limiting in more severe way crop production. Stress recognition in plants primarily occurs in the plasma membrane, modification of which is necessary to maintain homeostasis in response to it. It is known that lipid transport proteins (ns-LTPs) participate in modification of the lipidome of cell membranes. Representative of this group, ns-LTP2.8, may be involved in the reaction to abiotic stress of germinating barley plants by mediating the intracellular transport of hydrophobic particles, such as lipids, helping to maintain homeostasis. The ns-LTP2.8 protein was selected for analysis due to its ability to transport not only linear hydrophobic molecules but also compounds with a more complex spatial structure. Moreover, ns-LTP2.8 has been qualified as a member of pathogenesis-related proteins, which makes it particularly important in relation to its high allergenic potential. This paper demonstrates for the first time the influence of various abiotic stresses acting separately as well as in their combinations on the change in the *ns-LTP2.8* transcript, ns-LTP2.8 protein and total soluble protein content in the embryonal axes of germinating spring barley genotypes with different *ns-LTP2.8* allelic forms and stress tolerance. Tissue localization of *ns-LTP2.8* transcript as well as ns-LTP2.8 protein were also examined. Although the impact of abiotic stresses on the regulation of gene transcription and translation processes remains not fully recognized, in this work we managed to demonstrate different impact on applied stresses on the fundamental cellular processes in very little studied tissue of the embryonal axis of barley.

## 1. Introduction

Adverse environmental factors limit cereal production, restricting food supply to the growing human population. Unfortunately, cereal production is highly vulnerable to abiotic stresses. Drought, temperature, and salinity are the primary abiotic stresses that limit growth, yield, and grain quality [1]. Furthermore, these stresses act predominantly together, which further limits plant growth [2].

**Funding:** National Science Center Poland, DEC-2015/17/B/NZ9/01481. The funder had no role in study design, data collection and analysis, decision to publish, or preparation of the manuscript.

**Competing interests:** The authors have declared that no competing interests exist.

The stage of plant lifecycle which is the most susceptible to abiotic stress is germination. An undeveloped root cannot absorb water and nutrients from deeper parts of the soil. Moreover, the surface of arable land is exposed to strong desiccation (water evaporation); therefore, the salinity of the top layer of the ground increases. Temperature is also felt to a greater extent on the soil surface. Germinating embryos do not perform photosynthesis, thus plant survival under unfavorable conditions relies on the use of storage materials and results in the longest achievable root possible, if the intensity of stress permits grain to germinate at all [3,4].

Primary site for receiving stress signals in cells is the plasma membrane [5]. Abiotic stresses cause significant changes in lipid fluidity, which in turn, initiates stress signaling by affecting membrane-bound proteins and altering membrane lipids. This chain of events results in the expression of specific genes responsible for triggering adaptive mechanisms [6]. An increase in membrane permeability is caused by the degradation of lipids and proteins resulting from increased level of reactive oxygen species and oxidative stress [7]. Depending on the severity of stress, the degree of lipid peroxidation also increases [7,8]. Therefore, keeping membranes organized and functional is essential for coping with environmental cues.

Non-specific lipid transfer proteins (ns-LTPs) participate in maintaining homeostasis within cell membranes [9,10]. The ns-LTPs show lack of specificity for various phospholipids and other hydrophobic particles or even divalent cations [11]. In addition, ns-LTP2 are characterized by a higher lipid transfer activity, because they can bind not only molecules of linear lipids (as ns-LTP1), but also sterols making them particularly important transporters of cell membrane components which are exchanged under abiotic stress conditions [12,13].

Plant genomes contain different numbers of *ns-LTP* genes; for example, wheat has 156 identified sequences, *A. thaliana* has 49, rice contains 52, and barley has 70 [14,15]. Despite the differences in their sequences, they all exhibit similarity in spatial conformation and the presence of an eight cysteine motif (8CM) [16,17]. Because of their number and common features, a division into five groups was proposed based on gene and protein sequences, glycosyl-phosphatidylinositol (GPI) anchor attachment sites, the 8CM pattern, and intron presence and position [18]. However, they are often also divided into two main subgroups based on their molecular mass—ns-LTP1 and ns-LTP2 [19].

The proteome of mature barley seed is rich in pathogenesis-related (PR) proteins [20,21]. *PR* gene expression is regulated during plant development and in response to environmental stress [22,23]. PR proteins have increased resistance to proteolysis, high temperature, and maintain stability under low pH, in which most plant proteins are denatured [24]. These properties make PRs advantageous with respect to plant defense against stress; however, their high stability may contribute to induce allergenicity [25]. The ns-LTPs have been identified as a major human allergen, particularly ns-LTP2.8, which was predicted to be specific only to the aleurone layer of barley kernels [26]. To date, 17 PR families have been identified, group 14 includes ns-LTPs [26,27].

Currently, there is very limited experimental information regarding the ns-LTP2 protein family and their coding sequences in barley. Therefore, this study examined whether long-term abiotic stress conditions acting alone or in various their combinations could induce or influence the *ns-LTP2.8* gene expression level and its protein synthesis in vegetative tissues and the embryonal axis of selected barley genotypes. We also examined whether variability within the *ns-LTP2.8* sequence had an impact on the ns-LTP2.8 protein amino acid sequence. We examined the impact of different stress conditions on the change in the total soluble protein accumulation level in the barley embryonal axis, which is one of the ways of osmotic adjustment in plants growing in unfavorable environmental conditions.

## 2. Materials and methods

### 2. 1. Plant material

Two groups of spring barley genotypes (12 in total) were used.

i. A set of six homozygous spring barley (*H. vulgare* L.) lines and their parental genotypes with distinct allelic forms of the *ns-LTP2.8* gene characterized by Mikołajczak [28] were used. Briefly, the parental genotypes of the backcross lines, MPS37 (erect growth habit) and MPS106 (prostrate growth habit), were obtained by the single seed descent technique (SSD) [29] from the hybrids of cv. Maresi (two-rowed, semi-dwarf, German brewing cultivar) and Pomo (six-rowed, fodder cultivar). MPS37 (donor)×MPS106 (recurrent) were backcrossed up to $BC_6$ generation supported by single nucleotide polymorphism (SNP) genotyping of the among others, *ns-LTP2.8* allele variation. Based on this $BC_6$ lines MPW14/7, MPW14/9, MPW14/19, and MPW15/14 were selected. SNP genotyping indicated that MPW14/9 and MPW14/19 carry the allelic form of *ns-LTP2.8* gene as MPS37 (here referred as *ns-LTP2.8.a* allele) and the MPW14/7 allelic form of the *ns-LTP2.8* gene as MPS106 (here referred as *ns-LTP2.8.b* allele). Interestingly, MPW15/4 carries allelic form of the *ns-LTP2.8* gene as MPS106, but it is similar to MPS37 in terms of genetic background [28,30] (Table 1).

ii. A set of six recombinant inbred lines (RIL, $F_{10}$) were obtained by crossing the cv. Maresi (drought susceptible) and Syrian Cam/B1/CI08887/CI05761 breeding line (here referred as CamB1). CamB1 parental genotype was supplied to Dr A. Górny (Institute of Plant Genetics Polish Academy of Sciences, IPG PAS) by Drs S. Grando and S. Ceccarelli from ICARDA in Aleppo and selected for the analyses on the basis of multiple previous studies that examined well its physiological and genetic characteristics and confirmed its increased tolerance to reduced water and nutrient supply [31–35]. Moreover, recent analyses done by Kuczyńska [36] showed a different response at the lipidome level of the CamB1 line, which is associated with the accumulation of certain lipids classes under abiotic stresses. Stability analysis allowed us to select lines with inherited advantageous traits, such as earliness (CamB1) and semi-dwarfness (cv. Maresi) with stable grain yield during drought. Based on this, genotypes described by Mikołajczak [30]: MCam53, MCam75, MCam87 (stable yield), and MCam71 (extensive line, better grain yield under stress conditions) were selected to be used in this study (Table 1).

**Table 1. Analyzed plant material.**

| plant material | characteristics | | |
|---|---|---|---|
| Maresi | European brewing variety with reduced plant height, carrying the semi-dwarf *sdw1.d* (*denso*) gene, parental form for the BC and RIL lines | | |
| CamB1 | Syrian breeding line, parental form for RIL | | |
| MCam53 | RIL, a stable line in terms of yield under various environmental conditions | | |
| MCam71 | RIL, an extensive line yielding better under drought stress conditions compared to control conditions | | |
| MCam75 | RIL, a stable line in terms of yield under various environmental conditions | | |
| MCam87 | RIL, a stable line in terms of yield under various environmental conditions | | |
| MPS37 | DH line, donor form in backcrossing | allelic form of the *ns-LTP2* gene | *ns-LTP2.a* |
| MPS106 | DH line, recurrent form in backcrossing | | *ns-LTP2.b* |
| MPW14/9 | $BC_6$ line | | *ns-LTP2.a* |
| MPW14/19 | $BC_6$ line | | *ns-LTP2.a* |
| MPW14/7 | $BC_6$ line, the least similar to MPS37 | | *ns-LTP2.b* |
| MPW15/4 | $BC_6$ line, the most similar to MPS37 | | *ns-LTP2.b* |

## 2. 2. Stress application

To estimate the impact of abiotic stress conditions on the induction or change of the *ns-LTP2.8* expression level and translation of its mRNA in vegetative tissues as well as in those forming barley caryopsis, experiments were carried out both on mature plant tissues and on germinating grain tissues of the same genotypes under similar abiotic stresses in in-vitro conditions.

I.  Abiotic stress in germinating barley grains: undamaged caryopses without signs of fungal infection were selected and sterilized [37]. Experiment was carried out using sterile Petri dishes (10 cm diameter, 10 barley kernels per dish, six dishes per each condition and genotype) padded with Whatman paper [38]. Firstly, all caryopses were incubated at 22˚C on the dishes with sterile water (5 ml per dish) to unify the imbibition, thus germination process under stress conditions. After 24 h, appropriate stress factors were applied to the each set of the dishes as follow:

    1.  control (C) application of 10 ml sterile water per dish;

    2.  drought (D) application of 10 ml polyethylene glycol 6000 (PEG6000) solution to a final concentration of 20% per dish. These conditions, as reported by Hellal [39], significantly reduces percentage of sprouted grains while increasing content of soluble proteins in them to the greatest extent. Higher PEG concentrations inhibits germination of barley grain;

    3.  salinity (S) addition of 10 ml NaCl solution to a final concentration of 100 mM per dish. As reported by Kanbar and lEl Drussi [40] these conditions significantly limited germination and development of barley at the juvenile stage. Effect of 200 mM salt concentration was also analyzed here, but it inhibited the germination process of barley;

    4.  temperature (T) was applied as 30˚C (day) and 10˚C (night);

    5.  simultaneous application of each stress made it possible to obtain all combinations of them (drought with temperature (DT), salinity with temperature (ST), drought with salinity l(DS), and drought combined with salinity and temperature (DST)).

    Duration of stress ended when the seeds were considered germinated that is, when radicle reached length of approximately 2 mm [41,42]. Caryopses were transferred to a new sterile Petri dish lined with dry Whatman paper, the embryonal axes and aleurone layers (positive control) were dissected with sterile scalpel and placed in a sterile 0.5 ml Eppendorf tube. Samples were immediately placed in liquid nitrogen. No aleurone or storage material was collected with the embryonal axes. Samples of approximately 60 mg (± 5 mg), were stored at -70˚C for further analyses.

II. Application of abiotic stress conditions on mature plants was conducted in controlled greenhouse and phytotron conditions. Soil taken from a field at the IPG PAS was used as a substrate. Soil was sifted through a sieve with a mesh diameter 0.8 mm and then mixed with sand in a weight ratio of 7:2. 20 seeds per pot were sown, and after germination, the number of plants was reduced to 10. Before sowing, seeds were treated with a systemic fungicide Funaben (Synthos Agro). During the experiment, spraying against diseases was applied as necessary, mainly against powdery mildew (Amistar (Syngenta)) and aphids (Decis (Bayer)) according to the manufacturer's doses. All abiotic stresses were applied from the tillering stage (21 BBCH) for 14 days. BBCH scale is widely used to identify the phenological development stages of a plant [43]. Material for testing was collected at the 14th day of stress exposure. To set-up stress conditions, following procedures were used:

1. control (C) (greenhouse)—before filling the pots with substrate, its initial weight was determined in order to determine the humidity of the pot with substrate during the experiment. For control conditions optimal for barley growth 70% FWC (field water capacity) was maintained [44];

2. drought (D) (greenhouse) - 20% FWC was used as stress condition [44]. To maintain appropriate substrate moisture, measurements were carried out using the traditional weighing method and a hand-held FOM/mts device based on the reflectometric method. Using the FOM/mts device, three measurements were made within the experimental vase, and then the volume humidity results were averaged and converted into weight humidity. The methodology was developed as part of projects conducted at the Cereal Phenomics Department of the IPG PAS [36,45];

3. salinity (S)—(greenhouse)—tested genotypes were watered once with an aqueous solution of sodium chloride (NaCl) from above to obtain a final concentration of 250 mM·dm$^{-3}$ in the substrate [36];

4. temperature (T)—(phytotron): control conditions were characterized by a temperature of 22˚C during the day and 18˚C at night, air humidity of 50–60% and a photoperiod of 16/8 h, and in stress conditions—temperature of 30˚C during the day and 10˚C at night [36,46];

5. simultaneous application of each stress factor made it possible to obtain all combinations of analyzed stresses (DT, ST, DS and DST).

Samples (mature root (separated from the soil by intensive rinsing with distilled water for 5 seconds), mature leaf (three best-developed leaves from each plant), crown tissue, mature shoot, aleurone layer and embryonal axes) from both types of experiments were collected during the day time and immediately frozen in liquid nitrogen after sampling. Aleurone layers and embryonal axes were collected according to the modified protocol of Daneri-Castro and Roberts [47].

## 2. 3. Analysis of ns-LTP2.8 coding sequence

In order to estimate the sequence variability of the *ns-LTP2.8* gene allelic forms described by Mikołajczak [28], coding sequence of *ns-LTP2.8* (Ensembl Plants ID: HORVU.MOREX. r3.4HG0417270 (HORVU4Hr1G089500.1); GenBank ID X15257.1) of each analyzed genotype was multiplied and sequenced (LGC Genomics, Berlin, Germany). DNA samples were isolated from leaves of all analyzed genotypes using Wizard® Genomic DNA Purification Kit (100 mg/ isolation) according to the manufacturers protocol. DNA fragment was multiplied by PCR reaction (95˚C/3 min; [95˚C/20 s; 58˚C/20 s; 72˚C/30 s] × 30 cycles; 72˚C/7 min; 4˚C/∞) using Phusion™ High-Fidelity DNA Polymerase (Thermo Fisher Scientific) according to the manufacturers protocol (primers listed in S1a Table).

## 2. 4. Analysis of ns-LTP2.8 transcript level

To screen the *ns-LTP2.8* expression level, total RNA was isolated from all collected samples (60 mg of tissue/isolation) according to Ogrodowicz [48]. RNA was extracted using the RNeasy Mini Kit (QIAGEN, Germany) according to the manufacturer's protocol with on-column DNase treatment (QIAGEN, Germany). Additionally, isolates were treated with TURBO DNase (Thermo Fisher Scientific, Lithuania) according to the manufacturer's instructions to exclude trace contamination of samples with genomic DNA.

Single-stranded cDNA was synthesized from total RNA (150 ng/μl) using the iTaq Universal SYBR Green One-Step Kit (Bio-Rad) according to the manufacturer's protocol. Primer specificity was confirmed by sequencing the product of RT-qPCR reaction (S1 Fig) (Adam Mickiewicz University, Poznań, Poland). The first step in RT-qPCR reaction was reverse transcription (50˚C/10 min). Then, thermal cycling was performed with an initial step at 95˚C (60 s), followed by 39 cycles of denaturation at 95˚C (10 s) and primer annealing (30 s) (primers listed in S1b Table). Each run was followed by a melting curve analysis. The data were normalized using reference genes, and their stability was confirmed using a geNorm algorithm with CFX Maestro v2.0 software (Bio-Rad) [49,50]. The most stable reference genes, *EF1α* (GenBank ID: Z50789.1,) and *UBI* (GenBank ID: M60175.1,) were used (S2 Fig). The *Hvns-LTP2.8* primers were designed using BLASTN (plants.ensembl.org). Primer3 [51] was used to estimate physical properties of the primers and their potential to form homo- and/or heterodimers was evaluated using IDT OligoAnalyzer (Integrated DNA Technologies, USA). Oligos were synthesized by Merck (sigmaaldrich.com). For each RT-qPCR run, isolates from the negative (leaf), positive (aleurone layer), and no template controls (NTCs) were included. Changes in gene expression were calculated by CFX Maestro software [52]. To evaluate performance of each primer set, standard curves using 2-fold serial dilutions of the whole RNA isolates mixture (based on six technical replicates) were prepared (S3 Fig).

## 2. 5. Preparation of total soluble protein (TSP) extracts

Approximately 60 mg of isolated tissues was added to a chilled mortar filled with liquid nitrogen. After disruption, whole powder was transferred into a 1.5 ml Eppendorf tube and extraction buffer (PBS with 0,1% Tween®20) was added in 1:2 [mg:μl] ratio. The samples were vortexed thoroughly, incubated on ice for 10 min, and centrifuged (10 min., 4˚C, 14000 rpm). Protein concentration was determined using Quick Start™ Bradford Dye (Bio-Rad) according to the manufacturer protocol [53]. The absorbance was measured at 595 nm (Shimadzu, UV-1800) and converted to concentration using the calibration curve of bovine serum albumin (BSA) dilutions. The supernatants were transferred to fresh tubes, frozen in liquid nitrogen, and stored at -80˚C for further analysis [54,55].

## 2. 6. Detection of ns-LTP2.8 by western blot

Extracted proteins were used to screen presence of mature ns-LTP2.8 protein across all the tested barley tissues. Protein extracts (30 μg TSP/well) were denatured and separated by SDS-PAGE using a 6% stacking gel and 17% separating gel [47] in a Mini-PROTEAN Vertical Electrophoresis Cell (Bio-Rad). Proteins were transferred to a PVDF membrane (Immobilon®PSQ, Merck) by semidry protein transfer (55 min, 76 mA–parameters determined experimentally). Membrane was blocked for 1 h at RT in 5% skim milk in TBST (50 mM Tris, 150 mM NaCl, 0.05% Tween®20, pH 7.5). Primary polyclonal antibody specific for the antigen (16 amino acid fragment of the ns-LTP2.8 protein [GHYVSSPHARDTLNLC] according to UniProt ID: P20145) was custom designed (Agrisera, Sweden) and produced in rabbit. Control antibodies (i) anti-H3 (Agrisera, Cat. No. AS10 710) and (ii) anti-UBI (Agrisera, Cat. No. AS08 307) were selected due to their recognition of protein with similar size to the tested protein (inter alia allowing to establish conditions for example of the quality and efficiency of the electrotransfer). Membranes were incubated (1 h, RT) in 50 ml Falcon tubes with Tris-buffered saline (TBS) diluted primary antibodies: anti-ns-LTP2.8 (1:850) and anti-H3 and anti-UBI (1:20000). Membranes were incubated with secondary antibody, goat anti-rabbit IgG (H&L) HRP-conjugated (Agrisera, Cat. No. AS09 602) for 45 min in RT (diluted 1:150 000 in TBS). After each step, membranes were washed (3 × 5 min) in TBST (0.1% Tween®20). The

chemiluminescence-based reaction development was carried out in the darkroom using X-ray film (Kodak MXB) and ETA C 2.0 (Cyanagen) as a HRP substrate for 8-min exposure time.

### 2. 7. ELISA assay of ns-LTP2.8

The ELISA test was used to accurately estimate ns-LTP2.8 protein content in barley embryonal axes using the same antibodies as for the western blot analysis. Each assay step, except the first coating was preceded by three washes with PBST buffer (PBS with 0,05% Tween®20). Poly-Sorb® plates (Nunc-Immuno 96-well microplate) were coated overnight at 4°C with the protein sample extracts 10 × diluted in PBS or with the control extract with the antigen starting from 500 ng/ml and serially diluted to 7,81 ng/ml to make the calibration curve. Blocking of nonspecific interactions was done by adding 5% skim milk in PBS for 1h at RT and 300 rpm. Primary antibody, diluted 1:150 in PBS, 100 μl/well was incubated for 1 h at RT and 300 rpm, followed by the secondary anti-rabbit polyclonal antibody HRP-conjugated (Agrisera, Cat. No. AS09 602) diluted 1:45 000 in PBS (100 μl/well) for 1 h at RT and 300 rpm. TMB substrate was added to each well (100 μl) and incubated for 20 min at RT and 300 rpm. The reaction was stopped by adding 100 μl of 0.5 M $H_2SO_4$ and absorbance was measured at 450 nm using the Model 680 microplate reader (Bio-Rad). The ns-LTP2.8 protein was quantified [μg/g FW (fresh weight)] according to calibration curve using Microplate Manager Software v. 5.2.1 (Bio-Rad) preceded by subtraction of the values for negative control.

### 2. 8. Data analysis

Analysis of variance was carried out in a model containing fixed effects of genotype, stress variant, and interaction of them, on log transformed data. Multiple comparisons of means for genotypes under control conditions were performed using Fisher's Least Significant Difference method ($p < 0.05$). Significance of contrasts between stress variants and control for individual genotypes was tested by F test at $p < 0.05$ [56], with Bonferroni correction for the number of contrasts. Statistical analyses and graphs were made in the Genstat 19 program [57].

## 3. Results

### 3.1. ns-LTP2.8 transcript sequence and its level under optimal and stress conditions

The sequenced fragments of *ns-LTP2.8* protein coding sequences ($> 91\%$ sequence coverage) do not differ between the tested genotypes (Fig 1).

Presence of the *ns-LTP2.8* gene transcript has not been detected in any mature vegetative tissue of any tested genotypes either under control as well as under any of applied stress conditions. For aleurone and embryonal axis, significant changes in the level of *ns-LTP2.8* mRNA were identified in all tested genotypes (S4 Fig). The lowest *ns-LTP2.8* transcript content under optimal conditions was observed in MCam75, whereas the highest was found in MPW15/4 (*ns-LTP2.8.b*) (Fig 2).

On the other hand, MPS37 (*ns-LTP2.8.a*) exhibited much higher *ns-LTP2.8* transcript level compared with MPS106 (*ns-LTP2.8.b*), however, about half less than MPW15/4 (also *ns-LTP2.8.b*). A varied tendency was observed within the MPW lines. Genotypes carrying the *ns-LTP2.8.a* allele (MPW14/9, MPW14/19) showed reduced expression compared with its donor genotype (MPS37), whereas genotypes carrying the *ns-LTP2.8.b* allele (MPW14/7, MPW15/4) exhibited higher expression level compared with their donor genotype (MPS106) (Fig 2).

ANOVA for *ns-LTP2.8* transcript level revealed statistically significant effects of genotype, stress variant and of their interaction ($p < 0.001$). Analysis of contrasts showed that each of

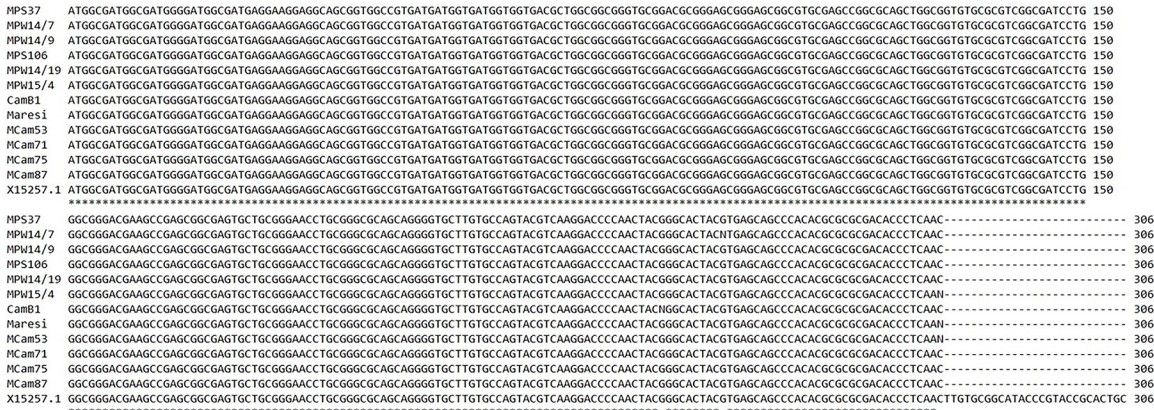

**Fig 1. ns-LTP2.8 protein coding sequence alignment (ClustalW, EMBL-EBI) of all tested barley genotypes and reference sequence (GenBank ID: X15257.1), analysis done in triplicate.**

the applied stress conditions resulted in statistically significant (F test, p < 0.001) increase of *ns-LTP2.8* transcript level compared with the control in barley embryonal axes (Fig 3) despite of the genotype. The single abiotic stress generally caused a greater increase in *ns-LTP2.8* transcript content compared with simultaneously applied stresses, except for combined drought and salinity, which resulted in the highest accumulation of *ns-LTP2.8* mRNA in all tested genotypes. The highest increase in transcript content induced by combination of drought and salinity stress was observed in MCam75 and MPS106 (*ns-LTP2.8.b*), whereas the lowest was in MPS37 (*ns-LTP2.8.a*).

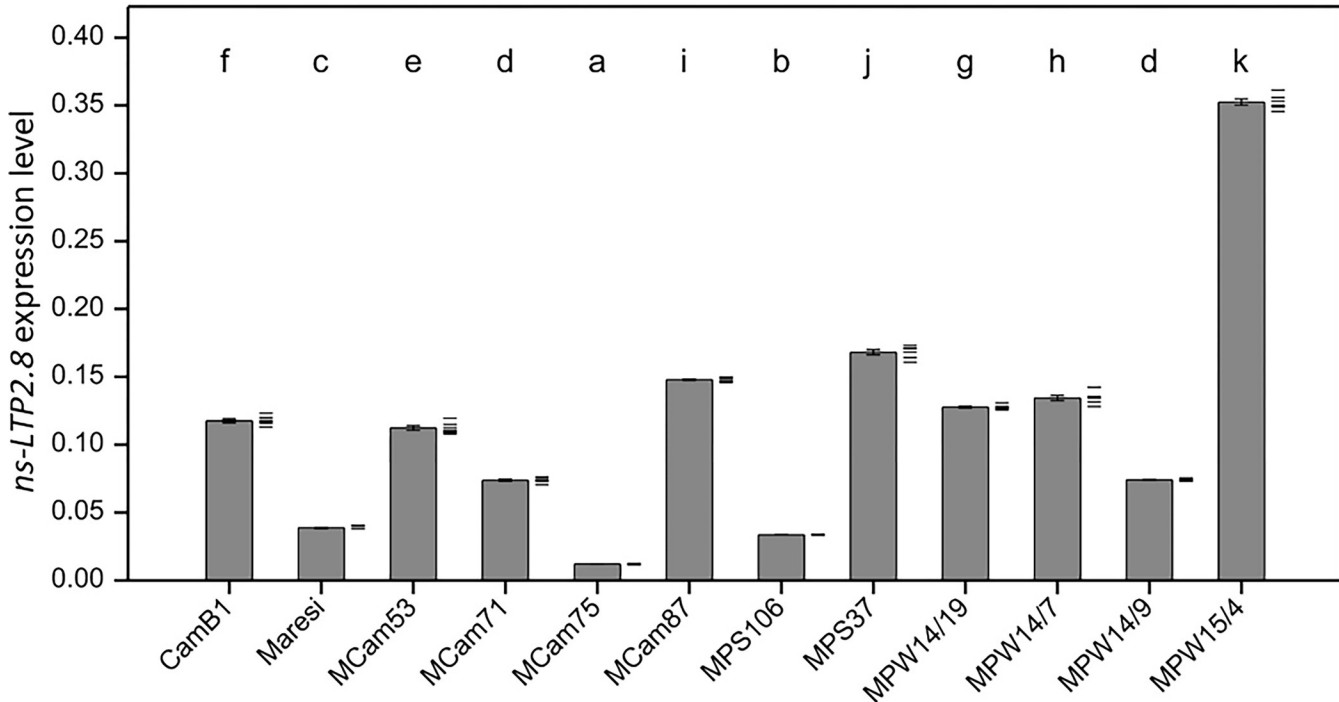

**Fig 2. Mean *ns-LTP2.8* expression level in all analyzed genotypes in control conditions.** Error bars represent standard error of the mean. Letters represent groups of similar genotypes (p < 0.05). *EF1α* and *UBI* served as a endogenous control for data normalization; six replicates were used for analysis, data points are shown as dashes next to bars.

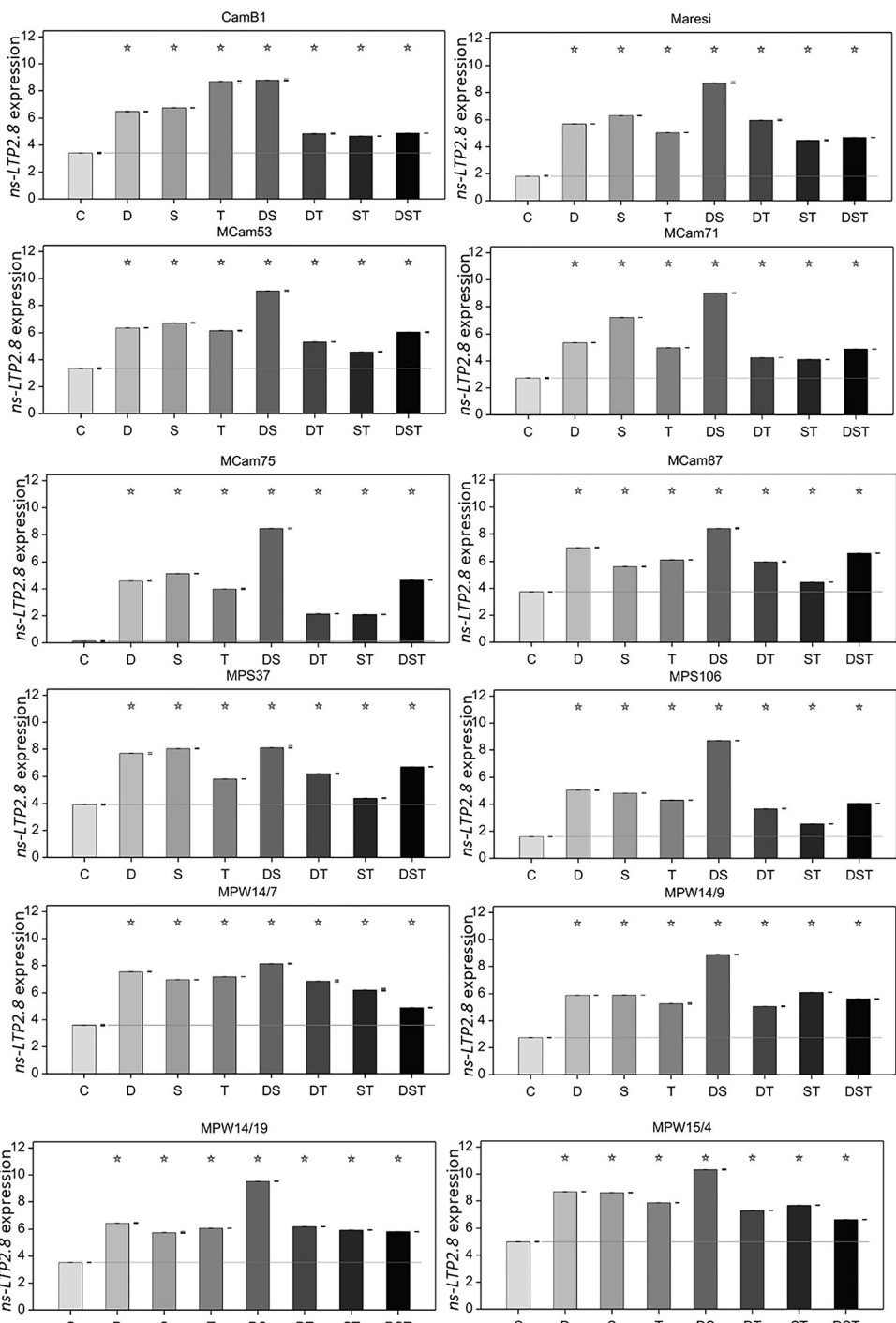

**Fig 3. *ns-LTP2.8* expression (in log₂ scale) under single and combined abiotic stresses in embryonal axis of different barley genotypes.** Error bars represent standard error of the mean values, stars denote mean values significantly different from control mean (F test, p < 0.001). *EF1α* and *UBI* served as a endogenous control for data normalization; six replicates were used for analysis, individual data points are shown as dashes next to bars.

Drought and salinity acting separately also significantly increased *ns-LTP2.8* transcript level, but this effect was much lower compared with exposure to their combination (Fig 3). The exception is the MPS37 genotype, within which a similar effect of salinity on the *ns-*

*LTP2.8* transcript level was demonstrated, similar to the effect of the drought and salinity combination. In turn, CamB1 showed a similar level of *ns-LTP2.8* expression under temperature stress to that detected in drought and salinity combination (Fig 3). In general, smallest increase was observed mainly in combined salinity and high temperature. Lines harboring monomorphic *ns-LTP2.8.b* allele exhibited similar pattern of change in *ns-LTP2.8* transcript accumulation, except for combined salinity and high temperature, in which they exhibited an increased mRNA synthesis compared with MPS106. The genotypes carrying the *ns-LTP2.8.a* allele exhibited more diverse relationship. The progeny lines did not show such a high impact of drought or salinity stress on *ns-LTP2.8* expression compared with the MPS37 parental line. Overall, besides the influence of combined drought and salinity, the highest level of *ns-LTP2.8* transcript level was observed under salinity stress in the most MCam lines and their parental genotype Maresi.

## 3.2. The influence of abiotic stresses on ns-LTP2.8 accumulation

Presence of the ns-LTP2.8 protein was confirmed by western blot analysis only in aleurone layer and barley embryonal axis both under control and all stress conditions and none of mature vegetative barley tissue showed its occurrence (S5 Fig). In general, increased TSP was noted after application of individual stresses, primarily salinity regardless of the genotype (Fig 4).

ANOVA for ns-LTP2.8 protein level revealed statistically significant effects of genotype, stress variant and of their interaction ($p < 0.001$). Analysis of contrasts showed that ns-LTP2.8 content decreased under abiotic stress conditions, applied either individually or simultaneously, with the exception of combined salinity and high temperature and combination of all three stresses (DST), which, regardless of a genotype, caused the greatest increase in ns-LTP2.8 concentration in relation to control conditions. In contrast, the lowest ns-LTP2.8 contents were observed in embryonal axes under combined drought and high temperature stresses for all genotypes. Similar effect was observed for TSP when a smaller effect on ns-LTP2.8 content was observed compared with the same stresses applied individually. A similar effect was observed under combined drought and salinity conditions for most MCam lines and their parental forms, whereas in most MPW lines, combination of drought and salinity conditions induced ns-LTP2.8 accumulation more than drought and less than salt stress. In addition, combination of drought and salinity affected protein content to a greater extent compared with the stresses applied individually in MPS37 (*ns-LTP2.8.a*), which is in contrast to MPS106 (*ns-LTP2.8.b*), where drought and salinity acting alone exhibited a greater effect than their combination (Fig 5).

## 3.3. Relationship between ns-LTP2.8 gene mRNA and ns-LTP2.8 protein content

Correlation analysis of contrast estimates for *ns-LTP2.8* mRNA and ns-LTP2.8 protein content in barley embryonal axis showed a considerable impact of abiotic stress or their combination on those variables relationship. A statistically significant, negative correlation was identified for salinity stress, while under the combination of all three stresses (DST) the correlation was positive. Based on the obtained data, it was found that among single stresses, only salt stress generated a negative correlation of *ns-LTP2.8* mRNA and protein content, while drought and temperature stresses generated positive ones, but not significant (Fig 6).

## 4. Discussion

According to the EMBL-EBI Expression Atlas, *ns-LTP2.8* gene transcription in the vegetative tissues of barley plants cultivated under optimal conditions has not been observed, however,

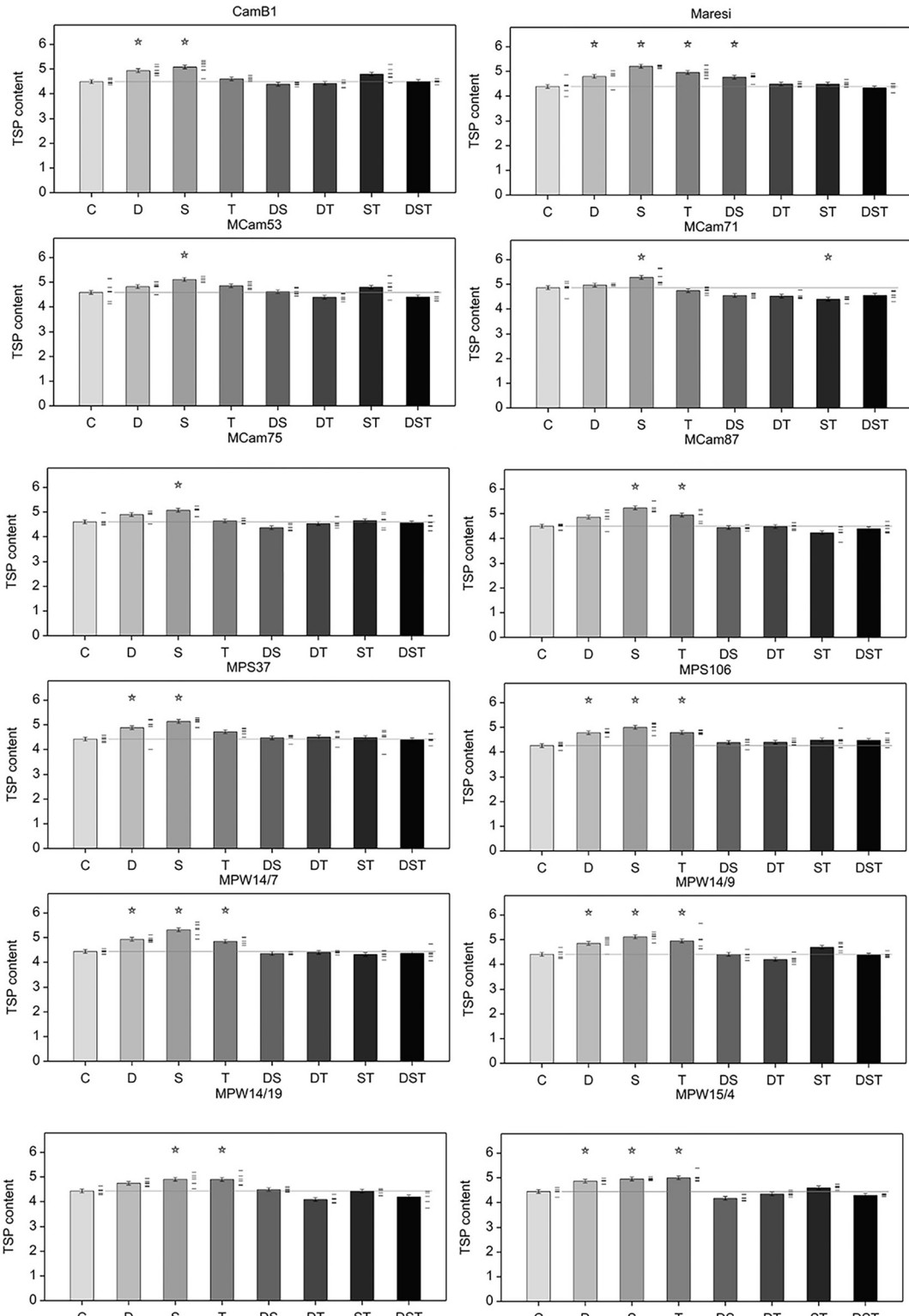

**Fig 4. TSP content (in log$_2$ scale) in barley embryonal axes in different abiotic stresses [mg/g FW].** Error bars represent standard error of the mean, stars denote mean values significantly different from control mean (F test, p < 0.001). Six replicates were used in analysis, individual data points are shown as dashes next to bars.

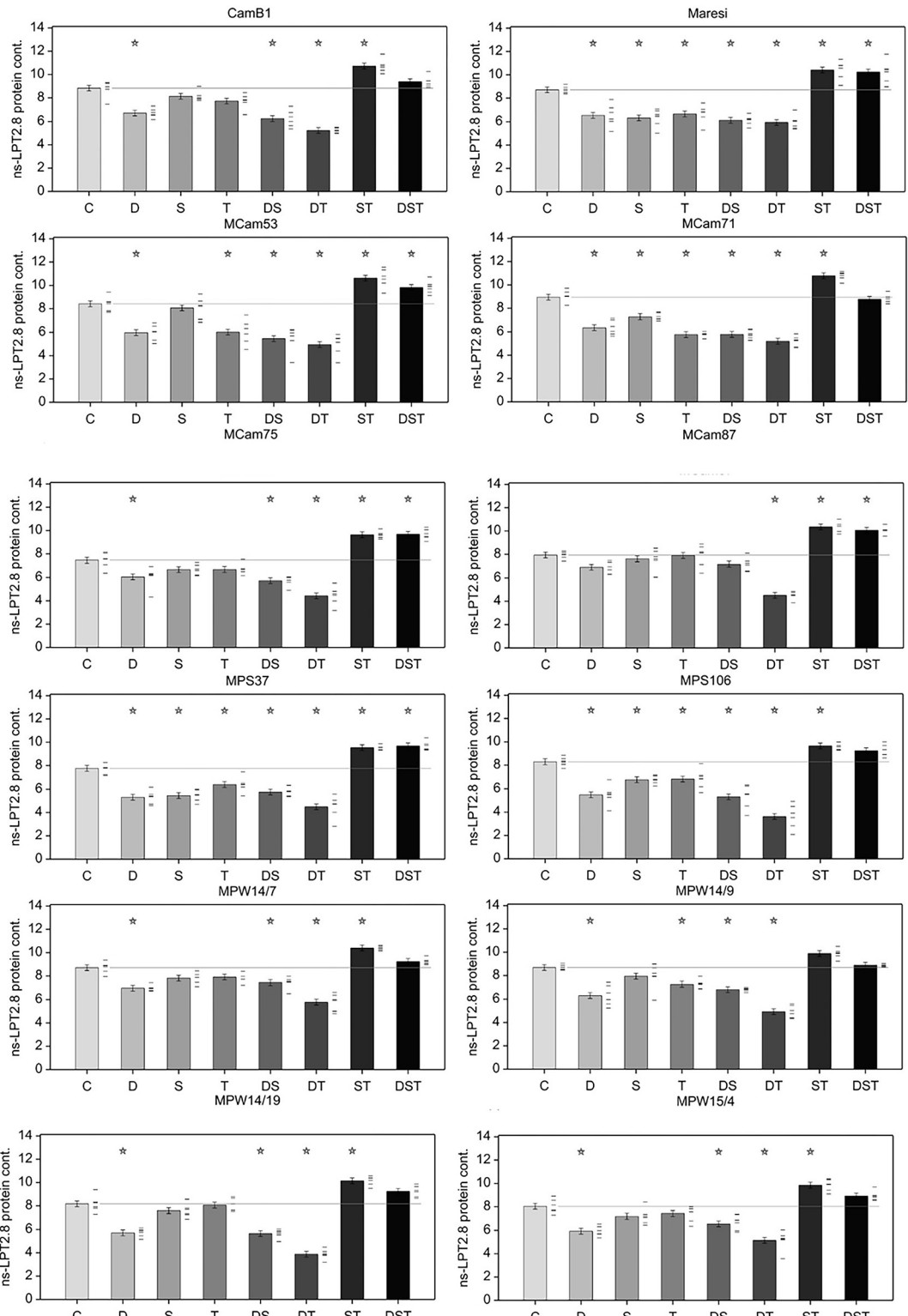

**Fig 5. ns-LTP2.8 protein content (in log$_2$ scale) in barley embryonal axes in different abiotic stresses [ng/mg TSP].** Error bars represent standard error of the mean, stars denote mean values significantly different from control mean (F test, p < 0.001). Six replicates were used in analysis; individual data points are shown as dashes next to bars.

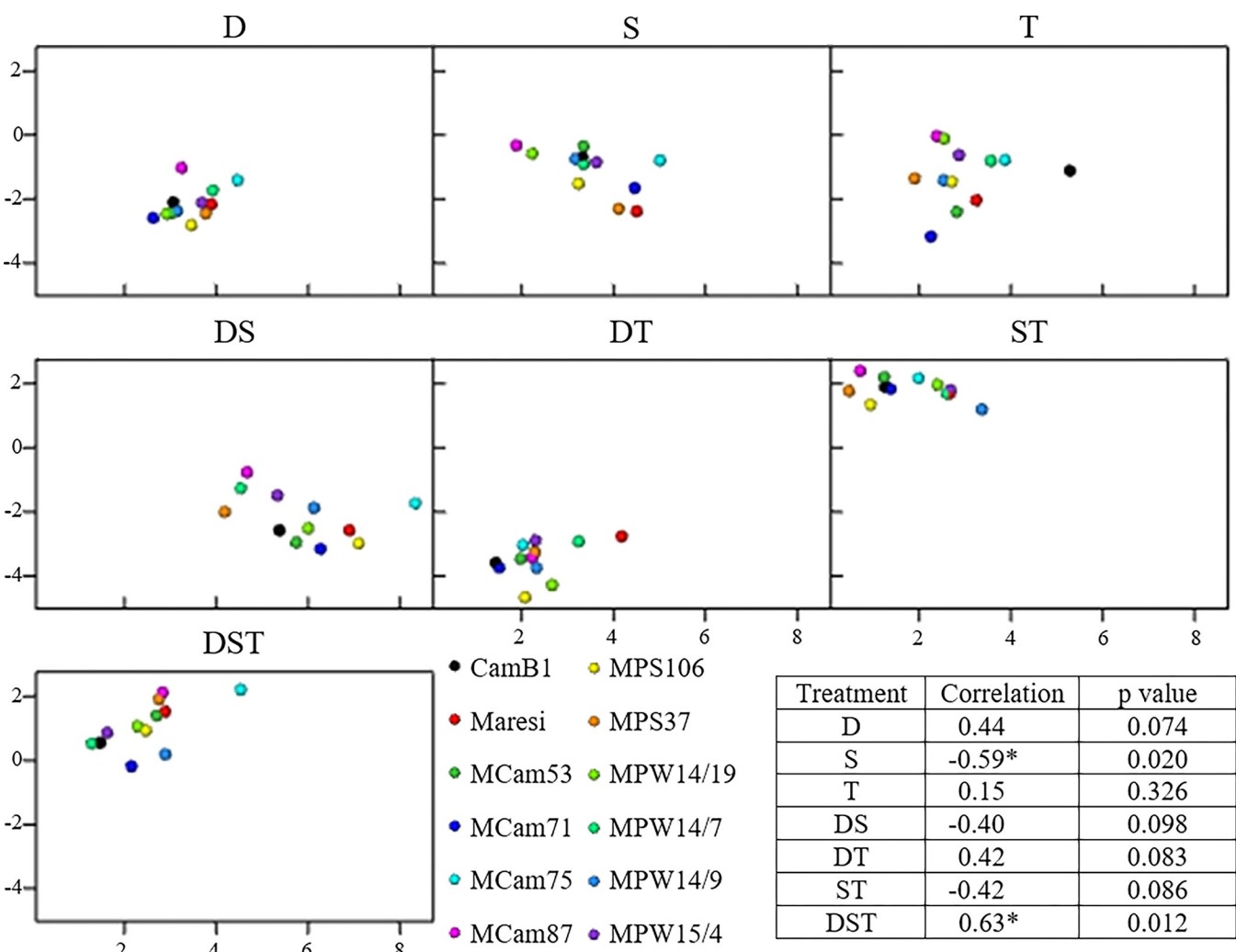

**Fig 6. Correlations between stress effects (stress v. control contrast estimate) for *ns-LTP2.8* mRNA content (x axis) and ns-LTP2.8 protein content (y axis).** Correlations significant at p < 0.05 are marked with stars.

the effect of severe abiotic stresses on the induction of its expression remained unknown. Duo [15] suggested that the expression of some genes encoding ns-LTPs may be induced by abiotic stress action. As a result of the experiments, we found that none of the most common naturally occurring abiotic stresses or their combinations induces its expression in vegetative tissues.

Barley *ns-LTP2* group consists of eight members [18], but each of them appears to be expressed in different tissues, for example Mikołajczak [58] identified three *ns-LTP2*-like genes (*ns-LTP2.4*, *ns-LTP2.5* and *ns-LTP2.6*) to be overexpressed in barley flag leaf exposed to drought and combined drought and heat stresses and Duo [15] mentioned about two *ns-LTP2* genes (ns-*LTP2.1* and *ns-LTP2.2*) to be expressed in roots. In studies done by Kalla [59] and Mayer [60], the presence of the *ns-LTP2.8* transcript was detected in the aleurone layer of developing barley kernels. The expression of ns-LTP2.8 encoding genes has been suggested to be specific only to this tissue [61]. However, in the embryonal axis, the expression of this gene and the change in its protein level induced by abiotic stresses has not been investigated so far. Although the function of ns-LTP2 proteins has not been fully described, the results presented in this work, combined with those obtained by others [15,58], may lead to the conclusion that

the expression of barley *ns-LTP2* genes is subject to tissue-specific regulation. Apart from the genes belonging to *ns-LTP2* expressed in the leaf or root, *ns-LTP2.8* is expressed and its protein synthesized not exclusively in the aleurone layer but also in embryonal axis of barley. This may lead to the conclusion that this protein participates in the nutrition of the germinating embryo by transporting hydrophobic molecules synthesized by the embryo or originating from the decomposition of reserve material contained in the grain and does not participate in the transport of hydrophobic ligands to any of the vegetative tissues. It was suggested that the presence of the ns-LTP2.8 protein in mature aleurone resulted from its stability after seed desiccation rather than its synthesis in mature tissue after inhibition [62,63]. Results of the present study confirmed the expression of the *ns-LTP2.8* gene in the aleurone of barley grain, but importantly, presence of *ns-LTP2.8* mRNA was also observed in the embryonal axis of barley as well as the effect of stress conditions was demonstrated for the first time. Based on our results, we claimed that apart from the function of this protein in transport within the cells of the aleurone layer the ns-LTP2.8 also participates in transport within cells of the developing embryo (or between those two cellular structures) to protect developing embryo from combined stresses (mainly combination of salinity and temperature). It is known that under constrained conditions proteins required for the stress response accumulate, whereas those not involved may be degraded [64] therefore, the role of the ns-LTP2.8 protein in abolishing the negative impact of these stress conditions in the barley embryonal axis seems to be significant.

The results obtained from SNP genotyping by Mikołajczak [28] suggested the existence of different sequences that encode the ns-LTP2.8 protein. However, sequencing of the protein coding sequence showed no differences between the sequences. The explanation for this might be the fact that there are possible differences in the promoter sequence of this gene, but not in the coding sequence. This statement is based on the fact that similar relationships have been noted in relation to the transcript level of this gene within the lines carrying its allelic forms—MPS37 and MPS106. Under control conditions, it was shown that the expression level between parental genotypes carrying separate *ns-LTP2.8* alleles is different. Moreover, genotypes carrying the *ns-LTP2.8.a* allele showed more similar average content of the *ns-LTP2.8* transcript under abiotic stress conditions compared to those carrying the *ns-LTP2.8.b* allele. In turn, the large difference in mRNA content of the *ns-LTP2.8.b* between the parental genotype (MPS106) and the MPW15/4 line may result from the fact that, according to Mikołajczak [28], the MPW15/4 genotype has the allelic form *ns-LTP2.8.b* consistent with the parental form MPS106, but the genetic background of MPW15/4 is most similar to MPS37, a genotype showing a higher expression level in control conditions. There are many papers stating about the fundamental influence of promoter sequences on the expression of many genes under abiotic stress conditions which affects binding of transcription factors and, therefore, modifying gene transcription level [65,66], hence the existence of differences between the promoters of the allelic forms analyzed in the study seems to explain the observed differences in the *ns-LTP2.8* mRNA content under the conditions of various abiotic stresses noted in the study.

In our study, both parental forms possessing various *ns-LTP2.8* gene allele exhibited different mRNA levels compared with their progeny lines. These results indicate the significant influence of mentioned genetic background of a given barley line on *ns-LTP2.8* expression which may additionally influence on the expression level of a given allelic forms of the *ns-LTP2.8* gene. SNP analyses revealed the presence of the *ns-LTP2.8.a* allele in the MPW14/9 line and the *ns-LTP2.8.b* allele in the MPW15/4 line, which showed greater deviations in the expression of *ns-LTP2.8* compared with their parental lines. This might also suggest the existence of differences in the promoter region of the allelic forms of this gene. Indeed, the vast variation in the promoter sequences of the *ns-LTP2* genes was documented by Duo [15], which may affect the efficiency of transcription factor binding, reflecting differences in gene

expression. Such a phenomenon is widely described in the literature for many transcription factors and proteins involved in regulation and/or transcription initiation [67]. Moreover, the existence of differences in the *ns-LTP2.8* promoter sequence and not the protein-coding sequence may be supported by the fact that a change in the coding sequence could contribute to a frameshift mutation, causing the synthesis of an incorrect protein or block its translation altogether. In the future, it would be necessary to sequence the promoter region of this gene in lines carrying different allelic forms of *ns-LTP2.8* and compare the results in search of differences in sequences that may contribute to different binding of transcription factors to it.

For an extended period of time, scientists have been analyzing the impact of single stress conditions generating changes in the physiology, genetics, proteomics and metabolomics of various plants. However, apart from knowing the impact of single stress, it is equally or even more important to estimate the impact of combined stresses (most often occurring in natural conditions) on the development and yielding of plants, especially in conditions of rapid population growth in the world. Prasch and Sonnewald [68] identified four types of plant responses to stress occurring simultaneously relative to those occurring individually, namely, a unique response, an additive effect or synergy of individual stresses and the dominance of one of them. They emphasized that the nature of this response depends on various factors, such as the intensity of the given stress and its duration. Our results indicate that a single stress condition increases *ns-LTP2.8* transcript level more significantly than combined stresses, with the exception of combined drought and salinity, in which the highest increase in *ns-LTP2.8* mRNA content was observed, regardless of genotype. This indicates an additive effect of these stresses on *ns-LTP2.8* expression level in barley embryos. In tobacco (*Nicotiana glauca*), at least one gene belonging to *ns-LTPs* was overexpressed during combined drought and salinity, which promoted the increased deposition of waxes on the outer side of the epidermis and is a nonspecific mechanism of stress protection [69,70]. Meanwhile, temperature stress applied simultaneously with salinity had the smallest mean effect on the *ns-LTP2.8* gene transcript level, suggesting a unique reaction in this stress variant on germinating barley plants. Correlation analysis showed that salinity is presumably a dominant stress among examined single stresses in relation to *ns-LTP2.8* mRNA and ns-LTP2.8 protein content in barley embryonal axis. This claim is supported by the fact that in DS and ST combination negative correlation has occurred even though drought and temperature stresses acting separately generated positive correlation values. This allows us to conclude that the effect of salt stress dominated over drought and temperature stress in relation to the *ns-LTP2.8* mRNA:protein ratio and only the interaction of drought and temperature with salinity was able to reverse this relationship.

Noteworthy, the highest *ns-LTP2.8* expression under combined drought and salinity conditions for all genotypes may suggest the existence of regulatory elements in its promoter that are responsible for the initiation of transcription under these certain conditions. It is known that co-occurrence of drought and salinity affects gene promoters that contain MYB family transcription factor binding sites [71–73]. Duo [15] showed that the promoter of the barley *ns-LTPs* genes contains multiple stress response elements, such as STRE, DRE, MBS, and TC repeats as well as sites related to hormonal signaling, such as ARE, LTR, ABRE, ERE, which are involved in stress response. The presence of so many regulatory motifs in the *ns-LTPs* promoters may explain the fact that in addition to combined drought and salinity, each of the stresses tested affected the expression of *ns-LTP2.8* in the different genotypes in a slightly different manner.

Abiotic stress induces a cascade of events in plant cells and affects expression of specific genes, which in turn, are responsible for the synthesis of specific proteins involved in adaptive mechanisms [6]. In the present study, single stress (mainly S) primarily increased TSP regardless of the genotype. This may be explained by an increase in the synthesis of proteins involved

in the response to abiotic stress, such as dehydrins, HSPs, transporters, or kinases in germinating barley embryos. In addition, a high cellular protein concentration represents a pathway of osmotic adjustment [74,75]. Considering these results, this type of response to salt stress appears to be a basic mechanism of protection in barley embryos germinating in high-salinity environments. On the other hand, the simultaneous action of various stresses can inhibit the course of energy-intensive cellular processes, such as translation [76–80], which may be basis of the negative relationships observed between *ns-LTP2.8* transcript and ns-LTP2.8 protein contents. An opposite relationship was observed for the ns-LTP2.8 protein content under combination of salinity and high temperature or combined all three stresses (DST) regardless of the genotype. These were the only conditions in which a positive relationship between transcription and translation of the *ns-LTP2.8* gene was observed. Therefore, their effect on the level of ns-LTP2.8 protein accumulation is additive. A similar relationship for combined salinity and high temperature stress was observed for some heat shock proteins in *Brassica juncea*, in which there was a significant increase in their expression and protein content compared with conditions in which these stresses occurred independently [81]. The demonstration of a higher ns-LTP2.8 accumulation level within the ST and DST conditions compared to the control may indicate an important role of ns-LTP2.8 in reaction to these factors in germinating barley.

An increase in the mRNA level of a given gene does not always correlate with its corresponding protein level [82]. Translational efficiency is affected by stress through activation or repression, resulting in changes in the mRNA:protein ratio [83,84]. We found that the highest increase in *ns-LTP2.8* mRNA level was under combined drought and salinity conditions, which did not concomitant with ns-LTP2.8 protein content. Presumably, translation was inhibited under combined drought and salinity conditions. This phenomenon has been described for other proteins, such as P5CDH and SRO5 in *A. thaliana* [85], ribosomal proteins in oat [86], and other proteins in both monocots and dicots [87]. While interpreting changes in the translation of *ns-LTP2.8* mRNA, the participation of initiation sequences within the 5' UTR of the sequence should be considered. The presence of upstream open reading frames (uORFs) in the 5'UTR mRNA may reduce translation of the main open reading frame (mORF) depending on environmental conditions. In addition to uORFs, other structures contained in the 5' UTR may affect the level of mRNA translation by interfering with it or acting preferentially on eukaryotic translation initiation factors (eIFs), ribosomal subunits, or other mRNA binding proteins. Involvement of this type of regulation has been reported for the *P5CR* gene in *A. thaliana* [88–90]. Another feature of the transcript encoding the ns-LTP2.8 protein is the high proportion of nucleotides that form a strong bond (~69%). In general, low G/C content characterizes highly translated mRNAs, whereas high G/C content is associated with repressed transcripts [91,92]. The high G/C content of the nucleotides in the 5'UTR as well as in the entire transcript may have a similar effect because of their propensity to form higher-order structures that block translation. For example, increased *PDH45* mRNA was observed under abiotic stress without a change in protein level [93–95]. In addition, *ns-LTP2.8* mRNA is short and the translation efficiency under abiotic stress is influenced by the length of the mRNA molecule. Under optimal conditions, the translation of short mRNAs is favored, whereas under stress conditions, long mRNAs (>2000 nt) are translated more efficiently [90,95–100]. The embryonal axis of barley is an intensively developing tissue that is rich in RNA. A significant amount of various transcripts in the embryonal axis suggests that regulatory molecules (ncRNAs: sRNA, siRNA, and miRNA) may be present in this pool in addition to coding sequences. Abiotic stress can induce or inhibit the expression of specific ncRNAs, thereby regulating the translation of other mRNAs in cells [101–104].

## 5. Conclusion

Presented work shows that none of the analyzed abiotic stresses or their combinations induce *ns-LTP2.8* gene expression in tissues other than aleurone and the embryonal axis in barley. It has been shown that each of the abiotic stresses significantly increases the *ns-LTP2.8* expression level and the conditions that most significantly increase its expression are the combination of drought and salinity regardless of the genotype. Conditions contributing most to increase the ns-LTP2.8 content in barley embryonal axis are conditions of salinity and temperature acting together despite the genotype. Moreover, we observed that the greatest increase in the total soluble protein content in barley, regardless of the genotype, was caused by salinity.

For future perspectives, we would recommend systematizing the names of the *ns-LTP* genes and, consequently, proteins. Currently there are several publications describing different number of ns-LTPs encoded by barley genome. Many studies describing this proteins base on *in silico* analyses of genomic sequences. In our opinion, this approach is not correct and in order to avoid the existence of several naming patterns for the same molecule, action should be taken. In addition, differences in the *ns-LTP2.8* expression level and its protein content in the embryonal axis could be more thoroughly investigated. It would be necessary to analyze the possibility of binding various proteins to the promoter sequence of this gene (for example by using EMSA technique), which would enable a better understanding of the mechanism of its regulation. Moreover, it would be interesting to perform polysome profiling in barley embryonal axes subjected to various types of abiotic stress in order to better approximate the mechanism underlying the regulation of translation under stress in germinating plants.

## Supporting information

**S1 Fig. *ns-LTP2.8* sequence amplified in RT-qPCR reaction, analysis done in triplicate.**
(DOCX)

**S2 Fig. Reference gene stability plot based on the built-in geNorm algorithm, calculated directly by the CFX Maestro software (Bio-Rad), analysis done in triplicate.**
(DOCX)

**S3 Fig.** Standard curves of *ns-LPT2.8* (a) and reference genes (b, c) used for analysis of *ns-LPT2.,8* mRNA content, calculated automatically by the CFX Maestro software (Bio-Rad).
(DOCX)

**S4 Fig. Relative *ns-LTP2.8* expression in different barley tissues under optimal conditions (purple) and salt stress (green).** Error bars represent standard error of the mean, analysis done in triplicate.
(DOCX)

**S5 Fig.** Detection of ns-LTP2.8 (exemplary western blot signals) in the embryonal axis and aleurone (three biological replicates, top panel) vs. reference proteins ubiquitin, H3 histone (two replicates) and negative control–mature organs (bottom panel).
(DOCX)

**S1 Table.** a. Primers used for sequencing of *ns-LTP2.8* coding sequence. b. Primers used for analysis of *ns-LTP2.8* mRNA content.
(DOCX)

## Author Contributions

**Conceptualization:** Michał Kempa, Krzysztof Mikołajczak, Piotr Ogrodowicz, Tomasz Pniewski, Paweł Krajewski, Anetta Kuczyńska.

**Data curation:** Michał Kempa, Tomasz Pniewski, Paweł Krajewski.

**Formal analysis:** Michał Kempa, Tomasz Pniewski, Paweł Krajewski.

**Funding acquisition:** Anetta Kuczyńska.

**Investigation:** Michał Kempa, Tomasz Pniewski.

**Methodology:** Michał Kempa, Tomasz Pniewski.

**Project administration:** Anetta Kuczyńska.

**Validation:** Michał Kempa.

**Visualization:** Michał Kempa, Paweł Krajewski.

**Writing – original draft:** Michał Kempa, Krzysztof Mikołajczak, Piotr Ogrodowicz, Tomasz Pniewski, Paweł Krajewski, Anetta Kuczyńska.

**Writing – review & editing:** Michał Kempa, Krzysztof Mikołajczak, Piotr Ogrodowicz, Tomasz Pniewski, Paweł Krajewski, Anetta Kuczyńska.

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
