## [Decision Letter · Decision Letter 0]

3 Dec 2023

PONE-D-23-33008The impact of multiple abiotic stresses on ns-LTP2.8 gene transcript and ns-LTP2.8 protein accumulation in germinating barley (Hordeum vulgare L.) embryosPLOS ONE

Dear Dr. Kuczyńska,

Thank you for submitting your manuscript to PLOS ONE. After careful consideration, we feel that it has merit but does not fully meet PLOS ONE’s publication criteria as it currently stands. Therefore, we invite you to submit a revised version of the manuscript that addresses the points raised during the review process.

We look forward to receiving your revised manuscript.

Kind regards,

Shailender Kumar Verma, Ph.D.

Academic Editor

PLOS ONE

Journal Requirements:

"National Science Center Poland, DEC-2015/17/B/NZ9/01481"

Reviewers' comments:

Reviewer's Responses to Questions

**Comments to the Author**

1. Is the manuscript technically sound, and do the data support the conclusions?

Reviewer #1: Yes

Reviewer #2: Partly

Reviewer #3: Partly

Reviewer #4: Yes

2. Has the statistical analysis been performed appropriately and rigorously? 

Reviewer #1: Yes

Reviewer #2: No

Reviewer #3: Yes

Reviewer #4: Yes

3. Have the authors made all data underlying the findings in their manuscript fully available?

Reviewer #1: Yes

Reviewer #2: Yes

Reviewer #3: Yes

Reviewer #4: Yes

4. Is the manuscript presented in an intelligible fashion and written in standard English?

Reviewer #1: Yes

Reviewer #2: Yes

Reviewer #3: Yes

Reviewer #4: Yes

5. Review Comments to the Author

Reviewer #1: The manuscript entitled “The impact of multiple abiotic stresses on ns-LTP2.8 gene transcript and ns-LTP2.8 protein accumulation in germinating barley (Hordeum vulgare L.) embryos” examined the impact of abiotic stress conditions either alone or in various combinations on ns-LTP2.8 gene expression level and its protein synthesis in vegetative tissues and embryonic axis of selected barley genotypes. However, there are few concerns which need to be addressed by authors before it can be considered for publication:

Comment 1 : Authors should conclude their study highlighting the major outcomes, significance as well as future prospects of the study . A separate conclusion section should be added.

Comment 2: As authors have performed real time PCR for expression analysis, The term RT-PCR used in the manuscript should be replaced with more appropriate term RT-qPCR.

Comment 3: Authors should mention in the manuscript why they have selected only one concentration of PEG and NaCl for application of drought and salinity stress? Have they screened for response to other concentrations?

Comment 4: Authors should mention in the figure legends of Figure 5, 6 and 7 which gene was used as housekeeping/ endogenous control for data normalization and how many replicates were used.

Comment 5: Legend of Fig 2 should be re-written as: Agarose gel photograph showing PCR amplification of ns-LTP2.8 gene from different barley genotypes. For each genotype, three replicates were used for PCR reaction. Red marks indicate saturated pixels in ChemiDoc™ XRS+ System (Bio-Rad). M represents 100 bp DNA ladder.

Reviewer #2: - The design of the experiment is relatively good. A lot of work has been done.

Comments:

- The manuscript needs English polishing

- The manuscript is very long and tedious for the reader. It is necessary to remove unnecessary parts and clean the remaining parts.

- The abstract of the manuscript can be written better. For example, authors should not provide an interpretation or conclusion about the gene before they describe their work. Also, the abstract is currently not a good representative of the study and does not have the impact it should have on the readers to encourage them to read the manuscript.

- The introduction is fragmented and suddenly jumps from one subject to another—for example, lines 31 to 34. Also, nsLTPs in plants and their role in plant development and response to environmental conditions have yet to be discussed. Most of the focus is on their allergenicity.

- Plant materials: good explanations are given. Is it possible for the authors to summarize the information in a table? Also, is there any information about the response to stresses for the first set of plants?

- How was the salt solution added to the pots? All at once or gradually? From above or below?

- How are the root samples separated from the soil?

- Which leaves were used for sampling?

- Figure 1 does not help to provide information.

- Figure 2 should be presented in the results. Also, please add negative and positive samples to the figure.

- Line 210-211 is repetitive. They were given in the previous section.

- Examination of gene expression: has DNA contamination been removed from RNA samples?

- Please provide the ANOVA results in a supplementary table. It is also better to compare the means with one of the more well-known tests (Duncan, LSD, etc.). If the interaction effect of the treatments is significant, the comparison of the means of the individual effects is not statistically correct.

I suggest additional statistical analyses (correlation, PCA, etc.). These analyses can show the relationship between the measured traits and the stress response. For example, has a relationship been between stress tolerance and gene expression or protein amount?

- About qPCR internal control genes: EF1a gene amplicon is longer than usual, and UB1 gene amplicon is shorter than usual. Please justify this.

- Figures related to gene expression analysis could be clearer. It needs to be clarified what the control is.

- As expected, the results have shown that the gene is not expressed in the vegetative tissues of the plant. I suggest the authors analyze this gene's promoter so that they can provide possible reasons for this lack of expression.

- But my most important question to the authors:

How can they prove with the presented data that the change in expression and amount of protein is effective in stress tolerance?

Reviewer #3: Summary: The data presented in this manuscript are part of a two important scientific problems: (1) abiotic stresses have major impacts on plant productivity and understanding how plants respond to stress is necessary to develop future crops in a warming world, and (2) accumulation of compounds for stress mediation may have unexpected consequences, i.e. increase in human allergens in stress-tolerant crops. While these are important problems to study, the scope of this manuscript is narrow – they measured transcript and protein levels for a single gene and gene product in 12 barley genotypes in response to eight stress conditions. While it is fairly narrow, this does represent a lot of work on behalf of the authors and is likely to be important for future research. I have a few recommendations for increasing the clarity and presentation of the data to improve the manuscript.

About barplots: in figures 5, 6, 7, 8, 9 and 10, and supplemental figure S3, results for multiple data points are presented in barplots with error bars. The authors should instead use a method to show the distinct data points, e.g. univariate scatter plots. For reasoning why I make this suggestion, please read “Beyond Bar and Line Graphs: Time for a New Data Presentation Paradigm” from Weissgerber et al 2015 (PLOS Biology 13(4): e100228, doi: 10.1371/journal.pbio.1002128).

Abstract: please better describe nsLTP-2.8, i.e. what it is and why it was chosen. The story that it is a food allergen that accumulates in grains and is a lipid transfer protein that plays a role in abiotic stress response is compelling. But as a reader, I did not understand these things until I read the introduction. This information should be made obvious in the abstract to better sell why this work is important.

Line 218: sequence data for RT-PCR product “data not shown”. Please show this in supplemental data.

Please provide an explanation or evidence that anti-nsLTP-2.8 antibodies do not bind other nsLTP-2 proteins. Particularly, the region in nsLTP-2.7 looks very similar – 11 out of 16 residues are identical, 3 out of 16 are strongly similar (residues 2, 4, and 5), and only 2 out of 16 (residues 14, 15) have no identity. Would the size of nsLTP-2.7 be notably different on the western blot gel?

Figure S3 represents the nsLTP-2.8 transcript level in all genotypes, aggregated into a single panel. These data should instead be shown individually – one panel for each genotype.

Reviewer #4: The paper was mainly about the influence of various abiotic stresses acting separately as well as in their combinations on the change in the ns-LTP2.8 transcript and ns-LTP2.8 protein contentsin the embryonal axes of germinating spring barley genotypes with different ns-LTP2.8 allelic forms and stress tolerance. I found it so much impressive and to innovative. However, I have some comments as below:

Abstract was soundly fine. The introduction chapter was organized and well justified.

l. 49-51, 66-67 requires citations; l. 71 authors are advised to spell out ‘GPI’;

Method section was well described, especially when they properly elaborated the approach on that basis they combined different stress factors (l.148-151). They have mentioned that stress endeded when radicle reached to the length of 2 mm. I suggest them add a citation here (l.152). In l.159 the expression "IPG PAS" possibly stand for a location or a study filed name: I recommend authors to clarify here. l. 167. a short description of BBCH sounds necessary here for readers. A spelling out for BSA and TBS terms in l. 247 and l.264, respectively is suggested. l. 336. the mentioned result requires evidence (by adding a Fig, number). Discussion chapter was properly explained their results with reasonable descriptions which was nicely associated with their findings and other results from earlier studies. Their suggestion for future studies was also impressive (l. 497-499). It can also be said that their combined stress factors could be considered as the innovative part of the study as it is most likely the case in natural conditions. In l. 549. the sentence edning with "ns-LTP2.8 gene was observed" is missing a full stop. A short conclusion for what was reported here seems necessary. Also, as my last suggestion, it could be much more interesting if they could explain a bit more about the relation between high temperature and drought stress factors in connection with salinity stress when they were explaining the opposite relationship ( not mandatory ).

6. PLOS authors have the option to publish the peer review history of their article (what does this mean?). If published, this will include your full peer review and any attached files.

Reviewer #1: No

Reviewer #2: No

Reviewer #3: No

Reviewer #4: **Yes: **Peiman Zandi

---

## [Author Response · Author response to Decision Letter 0]

18 Jan 2024

Reviewer #1:

Comment 1: Authors should conclude their study highlighting the major outcomes, significance as well as future prospects of the study. A separate conclusion section should be added.

The work was supplemented with a short conclusion (line: 562).

Comment 2: As authors have performed real time PCR for expression analysis, The term RT-PCR used in the manuscript should be replaced with more appropriate term RT-qPCR.

Appropriate changes in the manuscript body has been made (lines: 214, 215, 226, 865). 

Comment 3: Authors should mention in the manuscript why they have selected only one concentration of PEG and NaCl for application of drought and salinity stress? Have they screened for response to other concentrations?

We thank the Reviewer for noticing the lack of information. Changes in the manuscript has been made (lines: 137-140 and 142-144). In order not to increase the length of the manuscript, this has not been written, but we would like to clarify that during the preliminary analyses conducted as part of the presented research, an analysis of the impact of higher concentrations of both salt and PEG was performed. Increasing their concentrations significantly inhibited the germination of barley grains. It should also be taken into account that the work involved the simultaneous effect of abiotic stress conditions, which additionally prevented the use of higher concentrations of stress-causing factors, as this would inhibit grain germination (for example during the most complex DST conditions).

Comment 4: Authors should mention in the figure legends of Figure 5, 6 and 7 which gene was used as housekeeping/endogenous control for data normalization and how many replicates were used.

Changes in the captions of mentioned figures has been made. The figures were also renamed due to the exclusion of Fig 1, Fig 2a and Fig 2b from the manuscript.

Comment 5: Legend of Fig 2 should be re-written as: Agarose gel photograph showing PCR amplification of ns-LTP2.8 gene from different barley genotypes. For each genotype, three replicates were used for PCR reaction. Red marks indicate saturated pixels in ChemiDoc™ XRS+ System (Bio-Rad). M represents 100 bp DNA ladder.

Changes in the manuscript has been made and and if the Reviewer does not mind, in accordance with the suggestion to significantly shorten our publication, this photo will be removed. We believe that it does not provide significant information for the manuscript and was only intended to demonstrate the technical homogeneity of the bands being the product of the PCR reaction sent for sequencing. Importantly, the result of this part of the experiment is presented in the form of a sequence representation of the sequencing reactions (Fig. 1) (line: 293).

Reviewer #2:

Comment 1: The manuscript needs English polishing.

Copy editing was outsourced to an external company Enago. Enago certified their work as well as rate the manuscript as proficient. Additionally, the text of the manuscript was shortened and standardized.

Comment 2: The manuscript is very long and tedious for the reader. It is necessary to remove unnecessary parts and clean the remaining parts.

We have carefully reviewed our publication and have tried to remove fragments that do not impair its quality and readability. Our work was reduced by about 20%. 

Comment 3: The abstract of the manuscript can be written better. For example, authors should not provide an interpretation or conclusion about the gene before they describe their work. Also, the abstract is currently not a good representative of the study and does not have the impact it should have on the readers to encourage them to read the manuscript.

We are grateful for this valuable suggestion. The abstract has been significantly changed. All interpretations and conclusions have been removed from the abstract. 

Comment 4: The introduction is fragmented and suddenly jumps from one subject to another—for example, lines 31 to 34. Also, nsLTPs in plants and their role in plant development and response to environmental conditions have yet to be discussed. Most of the focus is on their allergenicity.

The introduction has been significantly changed and shortened.

Comment 5: Plant materials: good explanations are given. Is it possible for the authors to summarize the information in a table? Also, is there any information about the response to stresses for the first set of plants?

Table descripting plant materials has been added (Tab 1). Based on our previously research (Mikołajczak et al. 2016, 2017), we knew the response of the analysed genotypes only to drought conditions. This rich pool of genotypes was interesting in terms of their response to other abiotic stresses acting simultaneously - hence the purpose of the research. There are no studies in the available literature on the same genotypes under the conditions of all three stresses described in our publication.

Comment 6: How was the salt solution added to the pots? All at once or gradually? From above or below?

Answer to this question has been implemented in the manuscript (line: 180):

“salinity (S) - (greenhouse) - tested genotypes were watered once with an aqueous solution of sodium chloride (NaCl) from above to obtain a final concentration of 250 mM⋅dm-3 in the substrate (Kuczyńska et al., 2019)”

Comment 7: How are the root samples separated from the soil?

Answer to this question has been implemented in the manuscript (line: 189):

“mature root (separated from the soil by intensive rinsing with distilled water for 5 seconds)”

Comment 8: Which leaves were used for sampling?

Answer to this question has been implemented in the manuscript (line: 190):

“mature leaf (three best-developed leaves from each plant)”

Comment 9: Figure 1 does not help to provide information.

Thank you for this comment and we agree with the Reviewer Figure 1 has been deleted.

Comment 10: Figure 2 should be presented in the results. Also, please add negative and positive samples to the figure.

If the reviewer agrees, in accordance with the suggestion to significantly shorten our publication, this photo will be removed. We believe that it does not provide significant information for the manuscript and was only intended to demonstrate the technical homogeneity of the bands being the product of the PCR reaction sent for sequencing. Importantly, the result of this part of the experiment is presented in the form of a sequence representation of the sequencing reactions (Fig. 1) (line: 293).

Comment 11: Line 210-211 is repetitive. They were given in the previous section.

This issue has been resolved. The duplicated ns-LTP2.8 gene identifiers in lines 197-198 (left) and 210-211 (removed) have been corrected in this version of the manuscript.

Comment 12: Examination of gene expression: has DNA contamination been removed from RNA samples?

Yes, the RNA isolates were treated with DNase twice. The manuscript has been enriched with this explanation (lines: 207-211):

“RNA was extracted using the RNeasy Mini Kit (QIAGEN, Germany) according to the manufacturer’s protocol with on-column DNase treatment (QIAGEN, Germany). Additionally, isolates were treated with TURBO DNase (Thermo Fisher Scientific, Lithuania) according to the manufacturer’s instructions to exclude trace contamination of samples with genomic DNA.”

 It was decided to perform double DNase, because after performing only one (on column, Qiagen), products of genomic origin were detected using the NRT control (no reverse transcriptase control). Although products of genomic origin were detected around 30 - 32 cycle during the RT-qPCR reaction, after the second digestion of the genomic material, no increase in reaction products was detected in the NRT control in any of the samples 

Comment 13: Please provide the ANOVA results in a supplementary table. It is also better to compare the means with one of the more well-known tests (Duncan, LSD, etc.). If the interaction effect of the treatments is significant, the comparison of the means of the individual effects is not statistically correct.

Mean effects for all variation sources in ANOVA, for all traits, were significant at p < 0.001. This information is now included at the beginning of appropriate paragraphs (for transcript and protein content (lines: 312, 352). The advantage of the F test for comparisons is that it can be done in relation to the analysis of variance of all data for a particular trait, not separately for individual genotypes. We added in Methods information on the correction for simultaneous testing by Bonferroni method that was used (the significance threshold level is 0.05/84 = 0.001, approximately). We added a reference to the description of the F test for contrasts (Gomez & Gomez, 1984). As suggested in this comment, we removed analysis of marginal means for genotypes and treatments in section 3.3; now all analysis is done for genotype x treatment means, with an appropriate visualizations in Fig. 3, 4, 5. 

 Comment 14: I suggest additional statistical analyses (correlation, PCA, etc.). These analyses can show the relationship between the measured traits and the stress response. For example, has a relationship been between stress tolerance and gene expression or protein amount?

Analysis of the correlation between gene expression and protein content has been added in sec. 3.3. In Fig. 6, correlations are presented for different variants of stress (line: 380). 

Comment 15: About qPCR internal control genes: EF1a gene amplicon is longer than usual, and UB1 gene amplicon is shorter than usual. Please justify this.

Both reference genes used in this work were chosen from the publications (Rapacz et al., 2012; Al-Daoue et al., 2014) in which they were thoroughly analysed. This sets of primers have multiplied desired sequences and the product of RT-qPCR was homogenous. Those two reference genes were the most stable for performing RT-qPCR in our analysis and because of that fact they were used.

Comment 16: Figures related to gene expression analysis could be clearer. It needs to be clarified what the control is.

Reference genes names and number of replicates have been added to the picture descriptions (lines: 303-304, 325-326).

Comment 17: As expected, the results have shown that the gene is not expressed in the vegetative tissues of the plant. I suggest the authors analyse this gene's promoter so that they can provide possible reasons for this lack of expression.

Thank you for this comment. The promoter region of ns-LTP2.8 has been analysed and elements of aleurone-specific expression were presented by Kalla et al. (1994), however a more complete understanding of the promoter region of this gene using new molecular biology techniques could undoubtedly contribute to better understanding the mechanism regulating tissue-specific expression of this gene, especially in variable environmental conditions.

Comment 18: But my most important question to the authors: How can they prove with the presented data that the change in expression and amount of protein is effective in stress tolerance?

It is known that under stress conditions, the expression of genes and the translation of transcripts responsible for increasing the organism adaptation to environmental conditions are activated. It is known that genes not involved in plant adaptation do not significantly increase their expression under stress, and existing transcripts in cells do not have to be translated in response to stress. In this work an increase in transcript and protein levels of ns-LTP2.8 was demonstrated in the stress response. In addition, there are many publications stating that the remodelling of membrane lipids under abiotic stress is the basic response of cells to stressful conditions. The fact that hydrophobic molecules are transferred by proteins from the ns-LTP family links them directly with the lipid bilayer and the changes occurring therein. This study presents not the influence of ns-LTP2.8 on to lipid bilayer but the impact of various stress conditions and their combinations on changes in the level of gene expression and translation of the ns-LTP2.8 transcript. The barley genotypes examined in this study were analysed in terms of lipidomic changes caused by abiotic stress, as described by Kuczyńska et al., 2019. Presented publication could link changes in the lipidome with changes in the ns-LTP2.8 translation level, but this was not demonstrated in vegetative tissues. It is assumed that in mature tissues another protein from the ns-LTP family performs functions identical to those performed by ns-LTP2.8 in the aleurone/embryonal axis. 

Reviewer #3: 

Comment 1: About barplots: in figures 5, 6, 7, 8, 9 and 10, and supplemental figure S3, results for multiple data points are presented in barplots with error bars. The authors should instead use a method to show the distinct data points, e.g. univariate scatter plots. For reasoning why I make this suggestion, please read "Beyond Bar and Line Graphs: Time for a New Data Presentation Paradigm" from Weissgerber et al 2015 (PLOS Biology 13(4): e100228, doi: 10.1371/journal.pbio.1002128).

Thank you for this comment. We changed Fig. 2-5 to the form in which all data points are plotted next to bars. We also unified presentation for all traits in Fig. 3,4,5 to visualize both mean values and significance of comparisons to the mean value for control. 

Comment 2: Figure S3 represents the nsLTP-2.8 transcript level in all genotypes, aggregated into a single panel. These data should instead be shown individually – one panel for each genotype.

Suggested changes have been implemented. Due to the addition of Figure S1, name of Figure S3 has changed to Figure S4. Additionally, in accordance with the Reviewer, the previously combined expression level of the analysed gene in various tissues of all genotypes was divided into individual genotype per single panel.

Comment 3: Abstract: please better describe ns-LTP2.8, i.e. what it is and why it was chosen. The story that it is a food allergen that accumulates in grains and is a lipid transfer protein that plays a role in abiotic stress response is compelling. But as a reader, I did not understand these things until I read the introduction. This information should be made obvious in the abstract to better sell why this work is important.

We would like to thank the Reviewer for bringing this to our attention. Abstract has been changed.

Comment 4: Line 218: sequence data for RT-PCR product "data not shown". Please show this in supplemental data.

The lacking data has been added (S1 Figure).

Comment 5: Please provide an explanation or evidence that anti-ns-LTP2.8 antibodies do not bind other ns-LTP2 proteins. Particularly, the region in ns-LTP2.7 looks very similar – 11 out of 16 residues are identical, 3 out of 16 are strongly similar (residues 2, 4, and 5), and only 2 out of 16 (residues 14, 15) have no identity. Would the size of nsLTP-2.7 be notably different on the western blot gel?

Thank you very much for this substantive comment. The gene family encoding ns-LTP proteins is not fully known. Some of their representatives were selected using the in silico method, others were characterized at the transcript level, and still others were characterized at the protein level. It is known that the expression of genes encoding ns-LTP proteins is tissue-specific and some of these proteins are very similar to each other in terms of protein sequence. This situation was noticed by the reviewer in relation to the primary antibody. The ns-LTP2.8 gene is expressed exclusively in grain forming tissues (P20145, Expression Atlas, EBI) and ns-LTP2.7 gene apart from mentioned tissues is expressed in seedling (O81135, Expression Atlas, EBI). During preliminary research, a Western blot analysis of a protein isolates obtained from the above-ground part of a whole barley seedling was performed. However, unlike in positive control isolates obtained from the aleurone and embryonal axis, there was no band indicating the binding of the primary antibody to ns-LTP2.7. Hence, despite the high similarity of the ns-LTP2.7 and ns-LTP2.8 proteins, the primary antibody seems to be highly specific for the latter, as shown in the technical photo below. It can be added that the molecular weight of these two proteins differs by about 10%, which could be detected with a separation in a very dense gel and a sufficiently long electrophoretic separation. Moreover, ns-LTP2.7 and ns-LTP2.8 are characterized by the largest difference in the protein isoelectric point within the whole ns-LTP2 group. What is more, the Agrisera company was asked to produce an antibody recognizing a specific protein (P20145, UniProt), which is ns-LTP2.8, and the specificity towards the target was confirmed by this company

Fig. Technical screening of barley tissues in relation to ns-LTP2.8 protein presence. 

Reviewer #4: 

Comment 1: l. 49-51, 66-67 requires citations; l. 71 authors are advised to spell out 'GPI'. They have mentioned that stress ended when radicle reached to the length of 2 mm. I suggest them add a citation here (l.152). l.159 the expression "IPG PAS" possibly stand for a location or a study filed name: I recommend authors to clarify here. l. 167. a short description of BBCH sounds necessary here for readers. A spelling out for BSA and TBS terms in l. 247 and l.264, respectively is suggested. l. 336. the mentioned result requires evidence (by adding a Fig, number). In l. 549. the sentence ending with "ns-LTP2.8 gene was observed" is missing a full stop. A short conclusion for what was reported here seems necessary.

Above suggestions were followed and changes were made to the manuscript. former l. 49-51, 66-67 citations were given (lines: 50, 61-62); in former l. 71 'GPI' has been spell out (lines: 65-66). As we have mentioned, stress ended when radicle reached to the length of 2 mm. It has been suggest to add a citation here (former l.152), citation has been added (line: 150). It has been suggested to spell out (former l.159) the expression "IPG PAS" possibly stand for a location or a study filed name: I recommend authors to clarify here. The abbreviation “IPG PAS” was spell out earlier (lines: 108-109). In former l. 167. a short description of BBCH has been added (lines: 164-165). Abbreviations of BSA and TBS terms (in former l. 247 and l.264, respectively) was added (lines: 240, 257). In former l. 336. result was supplemented with evidence (by adding a Fig, number) (line: 328). In former l. 549. the sentence ending with "ns-LTP2.8 gene was observed" is missing a full stop – this issue has been resolved (lines: 517-519). A short conclusion for what was reported here seems necessary – this section has been added (line: 562).

Comment 2: Also, as my last suggestion, it could be much more interesting if they could explain a bit more about the relation between high temperature and drought stress factors in connection with salinity stress when they were explaining the opposite relationship (not mandatory).

The answer to this question may not be simple due to the huge differences in the effects of drought/salinity conditions and temperature stress. Drought or salinity stress is perceived mainly in the roots, while the root is least exposed to high temperature stress. The systemic response to drought and high temperature stress is opposite, a perfect example of which is the behaviour of stomatas, which are closed during drought to prevent greater evaporation of water from the leaves, while under temperature stress, stomatas are open to regulate plant temperature. A different response to drought/salinity/temperature stresses occurs not only at the physiological level but also at other fundamental levels such as transcription, translation and changes in the metabolome.

While revising your submission, please upload your figure files to the Preflight Analysis and Conversion Engine (PACE) digital diagnostic tool, https://pacev2.apexcovantage.com/. PACE helps ensure that figures meet PLOS requirements. To use PACE, you must first register as a user. Registration is free. Then, login and navigate to the UPLOAD tab, where you will find detailed instructions on how to use the tool. Please note that Supporting Information files do not need this step.

---

## [Decision Letter · Decision Letter 1]

9 Feb 2024

The impact of multiple abiotic stresses on ns-LTP2.8 gene transcript and ns-LTP2.8 protein accumulation in germinating barley (Hordeum vulgare L.) embryos

PONE-D-23-33008R1

Dear Dr. Kuczyńska,

We’re pleased to inform you that your manuscript has been judged scientifically suitable for publication and will be formally accepted for publication once it meets all outstanding technical requirements.

Kind regards,

Shailender Kumar Verma, Ph.D.

Academic Editor

PLOS ONE

Additional Editor Comments (optional):

Reviewers' comments:

Reviewer's Responses to Questions

**Comments to the Author**

1. If the authors have adequately addressed your comments raised in a previous round of review and you feel that this manuscript is now acceptable for publication, you may indicate that here to bypass the “Comments to the Author” section, enter your conflict of interest statement in the “Confidential to Editor” section, and submit your "Accept" recommendation.

Reviewer #1: All comments have been addressed

Reviewer #2: All comments have been addressed

Reviewer #3: All comments have been addressed

Reviewer #4: All comments have been addressed

2. Is the manuscript technically sound, and do the data support the conclusions?

Reviewer #1: Yes

Reviewer #2: Yes

Reviewer #3: Yes

Reviewer #4: Yes

3. Has the statistical analysis been performed appropriately and rigorously? 

Reviewer #1: Yes

Reviewer #2: Yes

Reviewer #3: Yes

Reviewer #4: Yes

4. Have the authors made all data underlying the findings in their manuscript fully available?

Reviewer #1: Yes

Reviewer #2: Yes

Reviewer #3: Yes

Reviewer #4: Yes

5. Is the manuscript presented in an intelligible fashion and written in standard English?

Reviewer #1: No

Reviewer #2: Yes

Reviewer #3: Yes

Reviewer #4: Yes

6. Review Comments to the Author

Reviewer #1: Authors have addressed all the concerns raised previously. However, authors need to take English language editing services as there are many grammatical errors in the entire manuscript. All those need to be corrected before manuscript can be accepted for publication.

Reviewer #2: (No Response)

Reviewer #3: Authors have adequately addressed the issues I previously raised. I appreciate the inclusion of the technical screening figure showing the seedling shoot western blot results, and I recommend that the reviewers include this image as a supplemental figure with a short description in the manuscript text to ensure readers have confidence in the authors’ antibodies. I also recommend providing the information about the differences of nsLTP-2.7 and nsLTP-2.8 molecular weight and localization in the manuscript text.

Reviewer #4: (No Response)

7. PLOS authors have the option to publish the peer review history of their article (what does this mean?). If published, this will include your full peer review and any attached files.

Reviewer #1: No

Reviewer #2: No

Reviewer #3: No

Reviewer #4: **Yes: **Peiman Zandi

---

## [Editor Report · Acceptance letter]

8 Mar 2024

PONE-D-23-33008R1 

PLOS ONE

Dear Dr. Kuczyńska, 

I'm pleased to inform you that your manuscript has been deemed suitable for publication in PLOS ONE. Congratulations! Your manuscript is now being handed over to our production team.

Kind regards, 

on behalf of

Dr. Shailender Kumar Verma 

Academic Editor

PLOS ONE